# Sequencing trait-associated mutations to clone wheat rust-resistance gene *YrNAM*

Fei Ni [1], Yanyan Zheng[1,5], Xiaoke Liu[1], Yang Yu [1], Guangqiang Zhang[1,2], Lynn Epstein [3], Xue Mao[1], Jingzheng Wu[1,6], Cuiling Yuan [1,7], Bo Lv[1], Haixia Yu[1], Jinlong Li[1,8], Qi Zhao [1], Qiyu Yang [1], Jiajun Liu[1], Juan Qi [1], Daolin Fu [1,4] ✉ & Jiajie Wu [1] ✉

Stripe (yellow) rust, caused by *Puccinia striiformis* f. sp. *tritici* (*Pst*), can significantly affect wheat production. Cloning resistance genes is critical for efficient and effective breeding of stripe rust resistant wheat cultivars. One resistance gene (*Yr10_{CG}*) underlying the *Pst* resistance locus *Yr10* has been cloned. However, following haplotype and linkage analyses indicate the presence of additional *Pst* resistance gene(s) underlying/near *Yr10* locus. Here, we report the cloning of the *Pst* resistance gene *YrNAM* in this region using the method of sequencing trait-associated mutations (STAM). *YrNAM* encodes a non-canonical resistance protein with a NAM domain and a ZnF-BED domain. We show that both domains are required for resistance. Transgenic wheat harboring *YrNAM* gene driven by its endogenous promoter confers resistance to stripe rust races CYR32 and CYR33. *YrNAM* is an ancient gene and present in wild wheat species *Aegilops longissima* and *Ae. sharonensis*; however, it is absent in most wheat cultivars, which indicates its breeding value.

Wheat stripe (yellow) rust, caused by *Puccinia striiformis* f. sp. *tritici* (*Pst*), can be a devastating wheat disease and poses a threat to global production[1]. Cloning resistance genes in wheat is essential for studying wheat-rust interactions and can expedite wheat breeding to combat stripe rust. However, of the 86 yellow rust (*Yr*) resistance loci reported in wheat, only seven *Yr* genes have been cloned[2], mostly through map-based cloning. *Yr10*, derived from Turkish wheat 'PI 178383', was transferred to wheat cultivars 'Moro'[3] and AvSYr10NIL[4], and confers all-stage resistance to *Pst*. Wheat cultivars with *Yr10* are resistant to multiple *Pst* strains, including CYR29, CYR31, CYR32, and CYR33, but are susceptible to CYR34[4–6]. Laroche and colleagues previously reported that a CC-NBS-LRR gene (Genbank AF149112) was a candidate for *Yr10*[7,8]. Their conclusion is based on experimental evidence that transformation of the *Pst*-susceptible cultivar 'Fielder' with AF149112

conferred resistance to *Pst* race SRC-84 but not to *Pst* race CDL-29, which is a fairly old race and equivalent to PSTv23 (synonym, PST-29) (according to personal communications with Dr. Xianming Chen). We previously performed haplotype and linkage analyses for AF149112, which we called *Yr10_{CG}*, and found that it was 1.2 cM away from the locus conferring resistance to mixed *Pst* spores of CYR29, CYR31, and CYR32[9,10]. Because transgenic wheat harboring AF149112 was susceptible to CYR29[10], a gene or locus other than *Yr10_{CG}* in the mapping region may confer resistance to CYR29.

Historically, resistance genes and other genes of interest were mostly cloned using a map-based cloning strategy, which requires genetic and physical maps and is tedious and time-consuming. But fine mapping around recombination suppression regions, where most alien resistance genes localize, is difficult. In contrast to map-based

---

[1]National Key Laboratory of Wheat Improvement, College of Agronomy, Shandong Agricultural University, Tai'an, Shandong 271018, China. [2]College of Agriculture and Bioengineering, Heze University, Heze, Shandong 274015, China. [3]Department of Plant Pathology, University of California, Davis, CA 95616, USA. [4]Spring Valley Agriscience Co., Ltd., Jinan, Shandong 250300, China. [5]Present address: Zhoucun District Agricultural Technology Service Center, Zibo, Shandong 255300, China. [6]Present address: Zhejiang Pharmaceutical University, Ningbo, Zhejiang 315000, China. [7]Present address: Shandong Peanut Research Institute, Qingdao, Shandong 266100, China. [8]Present address: College of Agronomy and Biotechnology, China Agricultural University, Beijing 100193, China. ✉e-mail: dlfu@sdau.edu.cn; jiajiewu@sdau.edu.cn

---

cloning, DNA and/or RNA sequencing-based methods use independent mutants to clone candidate genes. These methods target specific chromosomes or specific gene classes and thus avoid the interference of the rest of the genome. In addition, these methods do not require a genetic map. Recently, several rust-resistance genes have been cloned using sequencing-based methods. For example, *Yr7*, *Yr5/YrSP*, *Sr22*, *Sr26*, *Sr45*, and *Sr61* were cloned by MutRenSeq[11–13]; *Sr43* was cloned using the MutChromSeq method[14,15]; and *Sr62* was cloned by MutRNA-Seq[16]. However, each of these methods has different limitations. For example, MutRenSeq is only useful for nucleotide binding and leucine-rich repeat (NLR) gene cloning; MutChromSeq has a technically-demanding step of chromosome sorting by flow cytometry; and MutRNA-Seq requires a high-quality genome assembly of the wild-type (WT).

Here, we report the cloning of wheat stripe rust resistance gene *YrNAM* using the method of sequencing trait-associated mutations (STAM). STAM uses full-length isoform sequencing (Iso-Seq) of the WT as the reference and employs transcriptome sequencing of relevant mutants for gene cloning. We show that *YrNAM* encodes a protein with a NAM domain and a ZnF-BED domain. *YrNAM* is absent in most wheat cultivars but present in wild wheat species *Aegilops longissima* and *Ae. sharonensis*.

## Results

### STAM can clone genes of interest with mutants and RNA-Seq

To perform STAM, full-length WT transcripts are used to generate a reference of non-redundant transcripts. Then, RNA-Seq reads of loss-of-function mutants are mapped onto the reference, variants are called, and counts of mutated transcripts are tallied (Fig. 1a). In this study, a *Pst*-resistant WT P10−46[10] and seven *Pst*-susceptible $M_3$ lines (Fig. 1b) were used to perform STAM. Using the leaf transcriptome of the WT, we assembled 123,178 full-length non-concatemer (FLNC) isoforms, which contained our 40,564 non-redundant reference transcripts. After completing STAM, a total of 7434 mutation sites (G → A and C → T) were identified in the seven susceptible mutants. Only one transcript (T59230), was mutated in six of seven susceptible mutants (Table 1), i.e., all except M62. M62 segregated for *Pst* resistance; the *Pst*-susceptible segregant also has a G → A transition in T59230. Furthermore, T59230 is mutated in the other five susceptible mutants that were not submitted for RNA-Seq. Collectively, all 12 susceptible mutants have mutations in T59230 (Fig. 1c). Thus, T59230 is the candidate for stripe rust resistance.

### *YrNAM* is a stripe rust resistance gene with NAM and ZnF-BED domains

We performed DNA resequencing of P10−46 and de novo assembled ~1.1 M contigs of non-repetitive DNA with 1.8 Gb in total. Contig NODE_33935 (5633 bp, Supplementary Data 1) matches T59230 perfectly. T59230 CDS is 1227 bp in length and, by sequence alignment, is comprised of five exons (Fig. 1c). The predicted protein YrNAM has a NAM (No Apical Meristem) domain (*E*-value 2.7e−11) and a ZnF-BED (BED Zinc Finger) domain (*E*-value 2.5e−08) at the N and C termini, respectively. The YrNAM-GFP fusion protein is expressed in nuclei in tobacco leaves (Supplementary Fig. 1). Mutational analysis suggests that both NAM and ZnF-BED domains are required for resistance; six of the 12 loss of function EMS mutants have a mutation in the NAM domain, and the other six have one in the ZnF-BED domain (Fig. 1c).

### *YrNAM* co-segregates with the rust resistance in WT P10−46

To further validate that *YrNAM* confers stripe rust resistance in WT P10−46, we created an $F_2$ population using P10−46 and one *Pst*-susceptible mutant M19. The C2076T mutation of *YrNAM* in M19 was used to develop a unique diagnostic marker *YrNAM-F8R2* (*Hpy188*III digestion). As expected, 113 $F_2$ seedlings segregated as 3:1 for stripe rust resistance and susceptibility ($\chi^2 = 1.30$, $df = 1$, $P = 0.25$); *YrNAM-F8R2*

was completely linked to stripe rust susceptibility (Supplementary Data 2).

Previously, while re-mapping the *Yr10* region, we developed a small population of 126 $F_{2:3}$ plants[10] and a fine-mapping population of 7177 $F_{2:5}$ plants[10]. Among the 126 $F_{2:3}$ plants, the dominant PCR marker *YrNAM-F8R7*, which amplifies a 698 bp fragment of the 3′-end of *YrNAM*, completely links to the stripe rust resistance and the *Yr10* co-segregating marker *Xsdauw79* (Supplementary Data 3). Using 37 recombinants in the *Xpsp3000-Xsdauw79* interval obtained from the fine-mapping population[10], we showed that *YrNAM-F8R7* co-segregates with the stripe rust resistance derived from Moro (Supplementary Data 4).

### Transgenic *YrNAM* confers resistance to stripe rust

We constructed a native expression vector of *YrNAM* (PC1213) (Fig. 2a) and transformed it into Fielder and CB037, which are both susceptible to wheat stripe rust. Twenty $T_1$ seeds per line of three lines, including Fielder#3, CB037#4, and CB037#5, were tested with *Pst* race CYR32. The infection type (ITs) of each $T_1$ was correlated with the expression of *YrNAM* (Fig. 2b, c, Supplementary Data 5). The *YrNAM*-expressing $T_1$ plants in both Fielder and CB037 were immune (ITs = 0) or highly resistant (ITs = 1−2) to CYR32, while the *YrNAM* non-expressing $T_1$ and the negative control plants were highly susceptible (ITs = 7−8) (Fig. 2b, c). *YrNAM* expression from genetic complementation also conferred high resistance in two additional trials (Supplementary Fig. 2, Supplementary Data 5). In the $T_2$ generation, PC1213 transgenic plants showed resistance to CYR33, an avirulent race for *YrNAM* as documented in Moro, P8−13, and P10−46 (Fig. 3a). However, CYR34 is virulent to *YrNAM* because it caused abundant sporulation in Moro, P8−13, and P10−46 (Fig. 3b). Unexpectedly, PC1213 transgenic $T_2$ plants were resistant (ITs = 2) to CYR34 (Fig. 3b), which may have been caused by stacking *YrNAM* with *Yr6* in Fielder[17] and *Yr9* in CB037[18]. The WT Fielder is moderately resistant to CYR34, as indicated by a modest hypersensitive response (HR) (Fig. 3b).

### *YrNAM* is an ancient gene and is present in wild wheat species

In Chinese Spring chromosome 1B (IWGSC, RefSeq v2.1), *YrNAM* is homologous to *TraesCS1B03G0003600LC.1* and *TraesCS1B03G0003500LC.1*, which are adjacent to the *Yr10* diagnostic marker *Xsdauw79* (*TraesCS1B03G0003800*)[10] and around 1.0 Mb from *TraesCS1B03G0000200* (*AF149112*-corresponding gene) (Supplementary Fig. 3). The *YrNAM* homologs in chromosomes 1A and 1D were undetectable in RNA-Seq data from resistant WT P10−46, Moro, and the seven *YrNAM*-defective mutants. These homologs were only associated with low expression in public datasets (Supplementary Fig. 4). The phylogenetic analysis (Supplementary Figs. 5 and 6, Supplementary Data 6) and molecular marker investigation (Supplementary Fig. 7, Supplementary Data 7) revealed close homologs of *YrNAM* in *Aegilops longissima* and *Ae. sharonensis*, suggesting that the origin of *YrNAM* predates the speciation of wheat B- and D-subgenome progenitors, probably 7.3 million years ago[19]. The ZnF-BED domains were highly conserved among these *YrNAM* homologs, especially among genes annotated in chromosomes 1A, 1B, and 1S (Supplementary Fig. 8). In contrast, there was 37−44% identity between the YrNAM and the ZnF-BED domains of the six characterized NLR-BED genes for disease resistance in the *Poaceae* (Supplementary Fig. 8). The phylogenetic tree analysis showed the ZnF-BED domain of YrNAM is distinct from the ZnF-BED domains of NLR proteins in *Poaceae* (Supplementary Fig. 9, Supplementary Data 8). Thus, the ZnF-BED domain insertion in YrNAM is independent of the ones in NLRs Rph15 and Yr5/Yr7.

### *YrNAM* is absent in most wheat lines

To determine the distribution of *YrNAM* in cultivated wheat, we screened 60 *AF149112*-containing lines that were acquired from 659 wheat accessions[9,10]. Four lines (Fengmai 2, Fufan 24, Xianshixinmai,

and Xiangnong 3) were positive for *YrNAM-F8R7* (located at the 3′-end of *YrNAM*) but negative for *YrNAM-F1R3* (5′-end of *YrNAM*) and *YrNAM-FO2R2* (internal region of *YrNAM*) (Supplementary Fig. 10, Supplementary Data 9). We then screened another panel of wheat germplasm, including 578 Chinese cultivars and advanced breeding lines and 59 Hungarian cultivars; *YrNAM* was not identified in any of them (Supplementary Data 10). The general absence of *YrNAM* in most wheat might be caused by its close linkage to an undesired trait, the brown glume locus *Rg1*[20].

## Discussion

In this study, we used STAM, a simple and straightforward method for gene cloning in hexaploid wheat. It could be used in other plant species with complex genomes. STAM does not require a reference genome assembly; it only requires an Iso-Seq dataset from the WT and RNA-Seq data from multiple independent loss-of-function mutants that have a mutation in an exon. Our WT ($F_7$ generation) was homozygous, but in theory, STAM could be adapted for heterozygous plants. In addition, while we screened for EMS-type of mutations, the method could be useful for mutations such as small insertions or deletions generated by other mutagens.

**Table 1 | STAM results using Iso-Seq data from resistant WT P10–46 and RNA-Seq data from seven susceptible mutants**

| No. of non-redundant transcripts | No. of mutant lines that have variants in the indicated no. of non-redundant transcripts |
|---|---|
| 34,048 | 0 |
| 5680 | 1 |
| 760 | 2 |
| 72 | 3 |
| 3 | 4 |
| 0 | 5 |
| 1 | 6 |
| 0 | 7 |

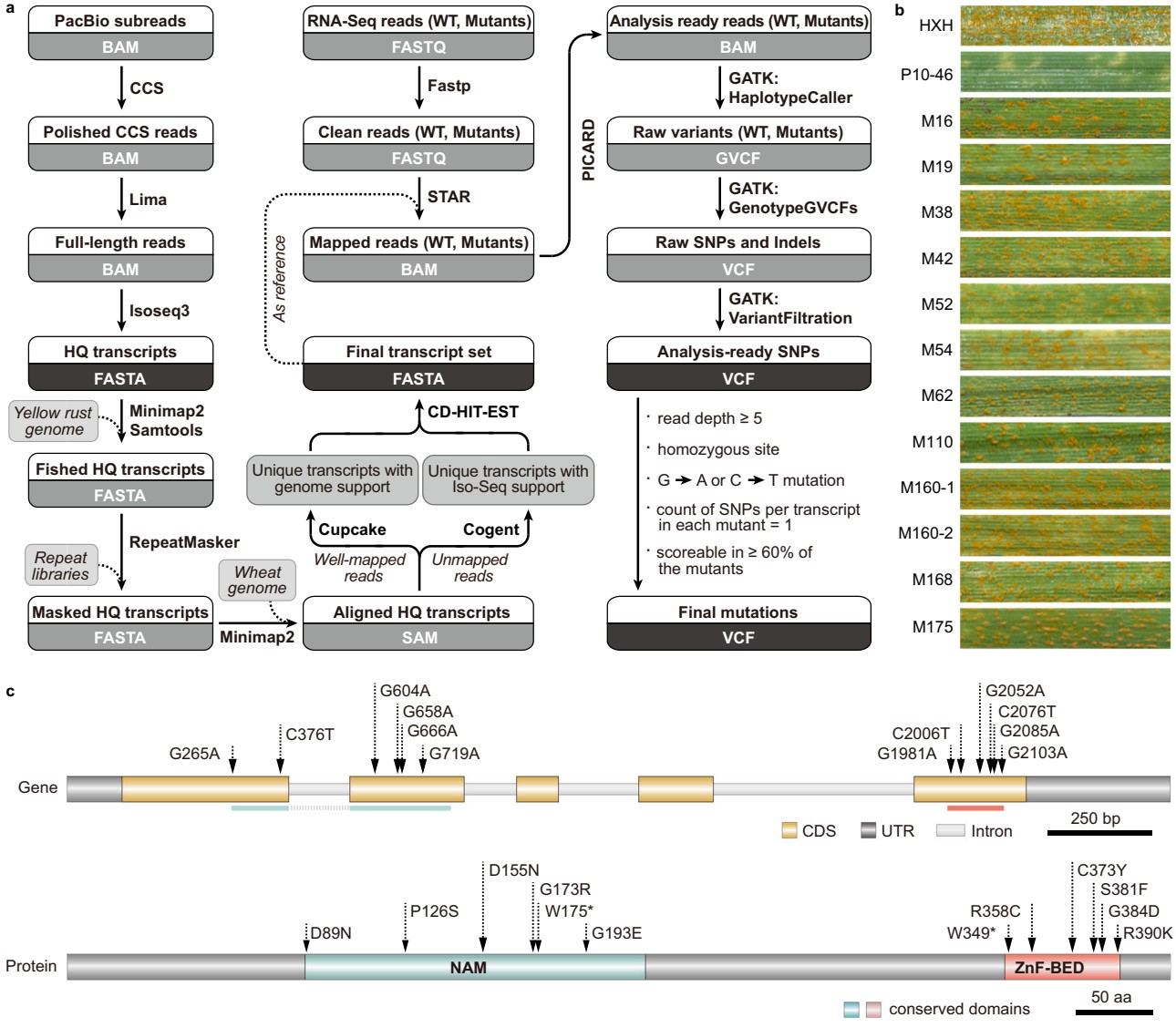

**Fig. 1 | Identification of *YrNAM* in wheat using STAM. a** Pipeline of STAM analysis. **b** Response to *Pst* (a mixture of races CYR29, CYR31, and CYR32) of the susceptible control cultivar 'Huixianhong' (HXH), the resistant wild-type P1046 with *Yr10*, and 12 *Pst*-susceptible mutants that were derived from EMS-treated P10–46. **c** Gene (1737 bp) and protein structure of *YrNAM* (T59230) and EMS-induced mutations in the NAM and ZnF-BED domains in the 12 mutant lines. The positions of the mutation sites are relative to 'ATG' or methionine.

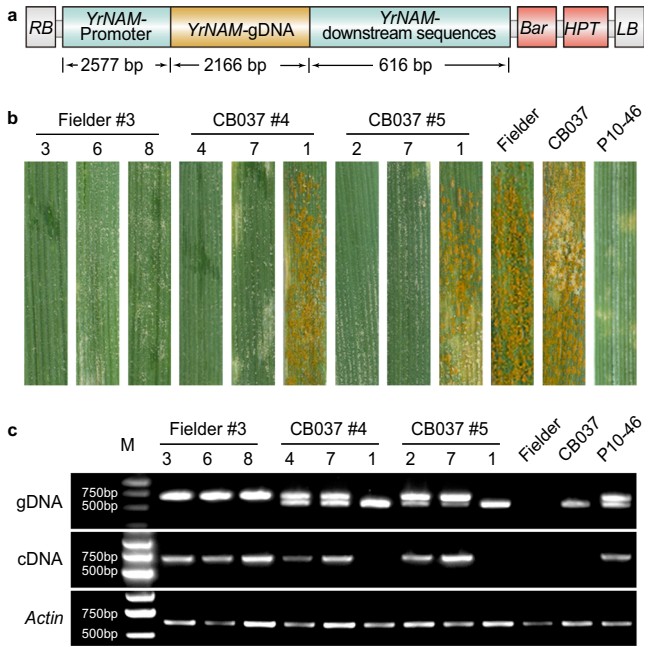

**Fig. 2 | Genetic complementation demonstrates that *YrNAM* confers stripe rust resistance. a** Native expression vector PC1213 was constructed for genetically transforming recipients with *YrNAM*. **b** Leaves with either a hypersensitive response (HR) or *Pst*-sporulation in resistant and susceptible genotypes, respectively. Stripe rust-susceptible Fielder and CB037 were transformed with *YrNAM*. Nine T$_1$ plants are shown on the left. P10−46 is the *Pst*-resistant control. Plants were inoculated with *Pst* race CYR32. **c** The integration (genomic DNA, or gDNA) and expression (cDNA) of *YrNAM*. Samples pair with those in (**b**). The target band for gDNA genotyping with primer pair *YrNAM-F8R7* is 698 bp; the smaller band (572 bp) in P10−46 and CB037 is amplified from the chromosome 1D homolog, which does not have *YrNAM*. *YrNAM-F4R43* was used for cDNA expression with a target band of 676 bp. The *Pst* test for T$_1$ plants using CYR32 was performed three times independently with similar results. Source data are provided as a Source Data file.

*YrNAM* is absent in most wheat cultivars, depriving the use of public genome assemblies to clone this gene as most of them are based on wheat cultivars. Alternatively, we can perform whole genome/chromosome sequencing of a WT with *YrNAM* to clone the gene. However, the cost of assembling a polyploidy genome like wheat is still high. Since STAM does not require whole genome/chromosome sequencing, it is a low-cost option for gene cloning. In addition, independent EMS mutants, which are required by STAM, are relatively easy to obtain in wheat, particularly when the trait is encoded by a single gene. Further, it is worth noting that the expression of *YrNAM* is low; we only detected 8 circular consensus sequencing (CCS) reads of *YrNAM* in 20 Gb Iso-Seq (~3.3 million CCS reads) from the WT, which translates to average 1.56−4.21 transcripts per million (TPM) in RNA-Seq of the WT at different time points after *Pst* inoculation. Thus, the success of cloning *YrNAM* indicates that STAM is feasible for cloning genes with low expression levels. However, STAM may not be a good option for cloning genes for polygenic traits. In addition, it is unsuitable for cloning genes with mutations in regulatory parts of a gene.

Recently, a method called MutIsoSeq, which is similar to STAM, was developed to clone wheat leaf rust resistance gene *Lr9/Lr58*[21]. In comparison to MutIsoSeq, which keeps multiple Iso-Seq isoforms derived from the same gene, STAM employs a non-redundant full-length transcriptome as a reference. In addition, the MutIsoSeq protocol aims to sequence about 70 Gb RNA-Seq data for each *Lr9/Lr58* mutant[21]; our study shows a smaller dataset of 15−18 Gb is sufficient for STAM. The success of cloning *Lr9/Lr58* and *YrNAM* indicates the effectiveness of using full-length transcripts instead of high-quality

genome assembly as the reference for gene cloning. STAM and MutIsoSeq add to the toolbox for gene cloning in wheat and other species with complex genomes. These methods are effective when target genes are either located in recombination-suppressed regions, absent in the reference genome, or encoding non-NLRs.

Till now, the *Yr10* region harbors two genes responsible for stripe rust resistance, *Yr10$_{CG}$* and *YrNAM*, which are separated by around 1.0 Mb based on the Chinese Spring reference genome (Supplementary Fig. 3). Since there is no reference-quality genome assembly of Moro or P10−46 available, the exact distance and sequence between *Yr10$_{CG}$* and *YrNAM* in these two accessions are unknown. In the functional validation of *Yr10$_{CG}$*[8], Moro plants that transfected with barley stripe mosaic virus (BSMV) constructs were susceptible to *Pst* race SRC-84; one transgenic line expressing *Yr10$_{CG}$* was resistant to SRC-84 but not to race CDL-29, which was detected in North America from 1983 to 1987 and was virulent on Moro[3]. Because *Pst* race CDL-29 was unavailable, we used Chinese yellow rust races (CYR), including CYR 29, CYR32, and CYR33, which are avirulent on Moro and WT P10−46, and CYR34, which is virulent on Moro and P10−46, for disease resistance assays. Transgenic Fielder and CB037 expressing *YrNAM* driven by its endogenous promoter are resistant to CYR 29, CYR32, CYR33, and CYR34. Without disease resistance assay using the historic North American strains such as CDL-29 that were used to characterize *Yr10*, we cannot determine whether *YrNAM* is the original *Yr10* or not. Regardless, the Wheat Gene Atlas[2] has called *Yr10$_{CG}$* a synonym for *Yr10*. To avoid confusion, we name the gene conferring resistance to *Pst* strains CYR29, CYR32, and CYR33 as *YrNAM*. We also note that *YrNAM* may hold some of the breeding value that was previously attributed to *Yr10* as it was considered the sole resistance in the region[22].

The dominant stripe rust resistance gene *YrNAM* encodes for two domains: 5' NAM and 3' ZnF-BED. Our mutational analysis suggests that both domains are required for stripe rust resistance. In alignments of *YrNAM* to available homolog sequences in public databases, the ZnF-BED domain is highly conserved (Supplementary Figs. 6 and 8), but the NAM domain is more divergent. Previously, the ZnF-BED domain was identified in several resistance genes to wheat stripe rust (*Yr5* and *Yr7*)[11], barley leaf rust (*Rph15*)[23], and the rice bacterial pathogen *Xanthomonas oryzae* pv. *oryzae* (*Xa1*)[24] and *X. oryzae* pv. *oryzicola* (*Xo1*)[25]. However, besides the ZnF-BED domain, these resistance genes contain an NLR domain, which is absent in *YrNAM*. While the ZnF-BED domain in NLR might act as a decoy for pathogen effector[11,26], the function of the ZnF-BED domain in the non-NLR YrNAM needs to be determined. Finally, we showed that *YrNAM* likely evolved before the speciation of wheat B- and D-subgenome progenitors. Since it is only present in a few wheat cultivars, there is potential for incorporating it into a wide variety of contemporary wheat cultivars for stripe rust-resistance breeding.

## Methods
### Plant materials
*Yr10* single-gene lines, P8−13 and P10−46, were derived from an F$_3$ population of a cross between *Pst*-resistant cultivar Moro and *Pst*-susceptible cultivar 'Huixianhong' (HXH)[10]. P8−13 and P10−46 are homozygous in the *Yr10* locus. The *Yr10* mapping population, containing 126 F$_{2:3}$ siblings of P8−13 and P10−46[10], was used for linkage analysis of *YrNAM* in this study. In a fine mapping effort working with 7177 F$_{2:5}$ plants, 62 homozygous and susceptible recombinants were obtained in the *Xpsp3000-Xsdauw79* interval[10]. Thirty-seven of the recombinants were also tested by PCR for *YrNAM*.

To determine the distribution of *YrNAM*, we tested a large collection of wheat germplasm: 60 *Yr10$_{CG}$* lines[9,10] (i.e., with Genbank accession AF149112, which, as shown here is approximately 1.0 Mb from *YrNAM*); 578 Chinese wheat cultivars (advanced lines); 59 Hungarian cultivars released between 1970−2013; and 56 *Ae. longissima*

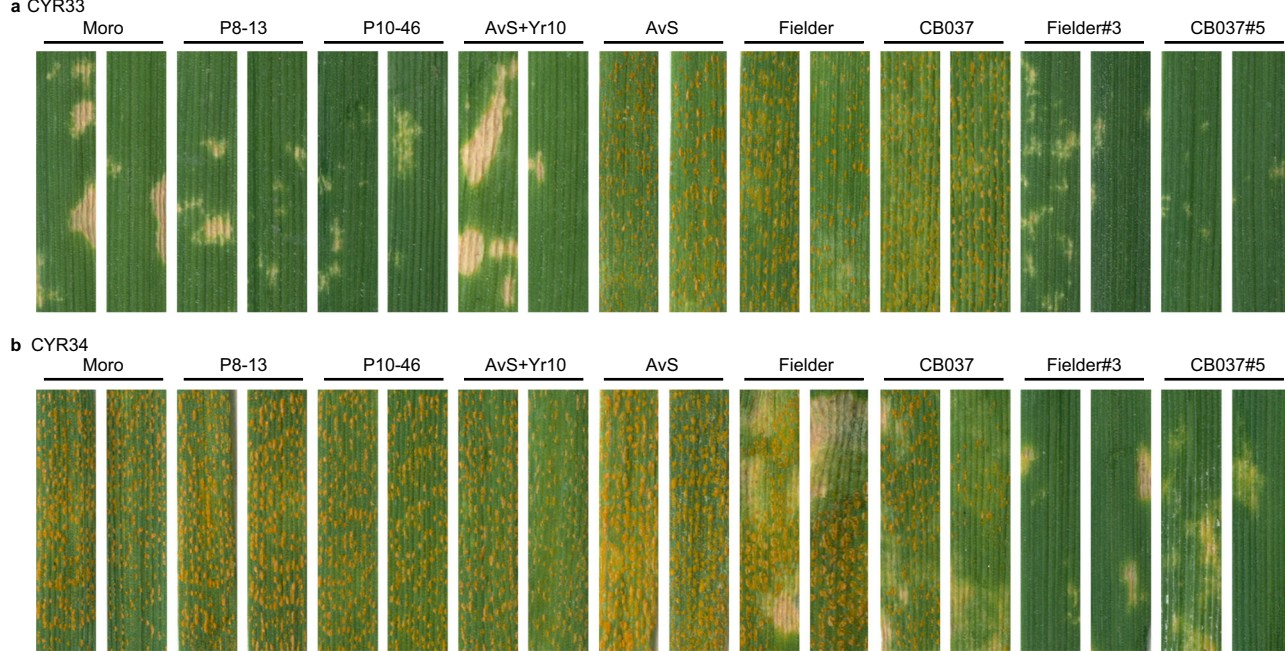

**Fig. 3 | Reaction types of *YrNAM*-containing materials to the avirulent and virulent *Pst* races. a** *YrNAM*-avirulent race CYR33. Moro, P8–13, P10–46, and AvSYr10NIL (AvS + *Yr10*), and *YrNAM*-transgenic Fielder#3 and CB037#5 are resistant with obvious HR (ITs = 2), while Avocet S (AvS) and the non-transformed Fielder and CB037 are susceptible (ITs = 7–9). **b** *YrNAM*-virulent race CYR34. Moro, P8–13, P10–46, AvSYr10NIL, and the Avocet S are highly susceptible (ITs = 7–9).

Wild-type Fielder and CB037 show moderate response (ITs = 4–6), characterized by obvious hypersensitive responses; however, *YrNAM* transgenic Fielder#3 and CB037#5 are highly resistant (ITs = 1–2) lacking sporulation. This experiment was performed once for CYR33 and three times for CYR34 independently, with similar results.

and *Ae. sharonensis* accessions from the U.S. National Plant Germplasm System.

## *Pst* races

A mixture of *Pst* spores that originally contained CYR29, CYR31, and CYR32 have been amplified in Taian, Shandong Agricultural University, and was used for field trials with the EMS mutants and growth chamber trials with the transgenic plants. The transgenic plants were additionally tested with the single race CYR32 and CYR33 provided by the Institute of Plant Protection, Chinese Academy of Agricultural Sciences, and CYR34 provided by the College of Plant Protection, Northwest A&F University, and a mixture of spores from naturally infected wheat fields in Luohe, Henan province, China (collected by Luohe Academy of Agricultural Sciences).

## EMS mutagenesis

In 2016, about 10,000 seeds of P8–13 and P10–46, which were homozygous in *Yr10*, were treated with EMS (Sigma-Aldrich Co., St. Louis, MO, USA). Both 1% (97 mM in water) and 1.05% (102 mM in water) were used to treat 5000 seeds. Each batch of 400 seeds (M$_0$) was soaked in 200 mL EMS solution in a 1-l flask, incubated on a shaker at 150 rpm at 25 °C for 10 h, and washed with running water at room temperature for 4 h; this protocol has a kill rate of approx. 75%. The processed seeds (M$_1$) were spread out on wet paper towels inside covered plastic boxes and germinated in a growth chamber at 25 °C and 16 h light. After one week, seedlings were transplanted into a greenhouse. A total of 2100 M$_1$ plants were harvested in June 2017. Every 10 M$_1$ plants were harvested as a pool, generating 211 pools (M1–M211). To identify *Pst*-susceptible mutants, 100 random M$_2$ seeds per pool were sowed in the field in Oct. 2017. In total, 14 segregating pools and 20 *Pst*-susceptible mutants were acquired. After confirming the *Yr10* region by *Xsdauw79* (a *Yr10* co-segregating marker)[10], we chose 12 *Pst*-susceptible mutants for further studies, including M16,

M19, M38, M42, M52, M54, M62, M110, M160-1, M160-2, M168, and M175.

## Sequencing trait-associated mutations (STAM)

For Iso-Seq of resistant WT P10–46, leaf tissues were sampled at 0 and 24 h after *Pst* infection. Total RNA was extracted using TRIzol reagent (Tiangen Biotech Co., Beijing, China) following the manufacturer's protocol. RNA integrity was evaluated by Agilent 2100 Bioanalyzer (Agilent Technologies, CA, USA). Two micrograms of total RNA were used for first-strand cDNA synthesis and PCR amplification. Two sequencing libraries (0 and 24 h after *Pst* infection) were prepared using the Clontech SMARTer PCR cDNA Synthesis Kit (Clontech, Dalian, China) and then sequenced using the PacBio Sequel II system (PacBio, CA, USA).

For whole-genome DNA resequencing of P10–46, DNA was isolated from uninoculated leaves using the standard CTAB method[27]. Genomic fragments (~250 bp) were prepared for Illumina NovaSeq (2 × 150 bp reads) and produced ~153 Gb raw data. Clean data were mapped to the *Poaceae* repeat element database mipsREdat 9.3p (PGSB Repeat Database) using BWA (version 0.7.17); unmapped reads (44.2 Gb, ~29%) were recovered using a homemade Perl script, and a de novo assembly was performed using SPADES v3.15 (https://github.com/ablab/spades).

Wheat cultivar Moro and seven *Pst*-susceptible mutants (M16, M19, M38, M62, M110, M160-1, and M168) were sampled for RNA-Seq. Total RNAs were extracted using TRIzol reagent from leaf tissues 24 h post *Pst* inoculation. Standard indexed libraries were prepared and pooled for Illumina sequencing, producing ~18 Gb paired-end data per library. Iso-Seq, DNA resequencing, and RNA-Seq were performed by Berry Genomics (Beijing, China).

The major steps of STAM are as below:

(1) De novo transcriptome reconstruction. The raw sequencing reads (PacBio subreads) were processed by ccs to generate polished

circular consensus sequence (CCS reads). After primer removal and demultiplexing using lima (version 2.2.0), IsoSeq3 (https://github.com/PacificBiosciences/IsoSeq) was used to trim PolyA tails and generate full-length, non-concatemer reads (FLNC reads); two SMRT cells FLNC reads were merged and clustered to generate polished transcript isoforms (HQ transcripts).

(2) Construction of final transcriptome reference. HQ transcripts were mapped to the genome reference of *Pst* isolate 134E[28] using Minimap2 (https://github.com/lh3/minimap2) to identify *Pst* contamination. After removing *Pst* transcripts, transposable elements were masked using RepeatMasker (http://www.repeatmasker.org) based on the repeat library Dfam3.6 (https://www.dfam.org). Afterward, clean HQ transcripts were mapped to wheat Chinese Spring genome reference IWGSC RefSeq v2.1[29] using Minimap2 in order to produce aligned HQ transcripts. To collapse redundant isoforms caused by 5′ RNA degradation, Cupcake and Cogent (https://github.com/Magdoll) were used to generate unique transcripts for well-mapped reads and unmapped reads, respectively. Then, the two non-redundant transcript data were combined as a final transcript set using CD-HIT-EST with the sequence identity threshold at 100% (-c 1), resulting in 40,564 non-redundant transcripts that were used as references for variant calling.

(3) Variant calling and rapid identification of a gene that confers a trait of interest. For RNA-Seq reads of Moro and *Pst*-susceptible mutants, adapter trimming, and quality filtration were performed using fastp (https://github.com/OpenGene/fastp); clean data was then mapped to transcript references from the previous step using STAR (https://github.com/alexdobin/STAR). Potential PCR-duplicated reads were also removed using Picard (https://broadinstitute.github.io/picard). SNPs were identified by the HaplotypeCaller tool of GATK v4.2 in GVCF mode (https://gatk.broadinstitute.org). Then, all the per-sample GVCFs were gathered and passed to GATK GenotypeGVCFs for joint calling. Variants were preliminarily filtered using GATK VariantFiltration with the parameter "$DP < 5 || FS > 60.0 || MQ < 40.0 || QD < 2.0$", which generated analysis-ready SNPs. Qualified SNPs met all the following conditions: (a) read depth $\geq 5$; (b) homozygous site; (c) either a $G \rightarrow A$ or a $C \rightarrow T$ mutation; (d) the count of SNPs per transcript in each mutant = 1; and (e) scoreable in >60% (four of seven) of the mutants. Amongst all seven mutant lines, a total of 6516 reference transcripts were identified with one or more $G \rightarrow A$ and/or $C \rightarrow T$ mutations.

## Genetic transformation of wheat

To construct a vector to express *YrNAM* with its endogenous promoter, we amplified a 5358 bp genomic fragment, which contains 2577 bp before 'ATG' to 616 bp after 'TAA' in contig NODE_33935, using DNA from P10–46, high-fidelity polymerase Phanta Max (Vazyme Biotech, Nanjing, China) and PCR primers F09 and OL-R3. The resultant fragment was cloned into the pENTR vector using a pBM27 toposmart cloning kit (Biomed, Beijing, China). Using LR Clonase™ II (Invitrogen, Waltham, USA), the *YrNAM* native gene was then cloned into PC613, which has 35S::*Bar* for plant selection. The final construct PC1213 was transformed into *Agrobacterium* strain EHA105. Genetic transformation of common wheat was carried out by *Agrobacterium*-mediated transfer[30]. Two wheat genotypes susceptible to stripe rust, Fielder and CB037, were used as recipients. Plants were grown in a growth chamber with 16 h light at 24 °C and 8 h darkness at 18 °C. $T_0$ transgenic plants were genotyped by amplifying *Bar* (Bar-F/R) and *YrNAM* (F8/R7).

## *Pst* inoculation

Moro, P10-46, HXH, Fielder, CB037, and EMS mutants ($M_2$, $M_3$), transgenic plants ($T_1$ and $T_2$), and selected recombinants ($F_{2:5}$) were tested for seedling resistance to *Pst* races. $M_2$ EMS mutants were tested in the field with a mixture of spores of CYR29, CYR31, and CYR32. In total, 211 pools (10 $M_2$ lines per pool) were planted as pool-based rows in fall 2017 and were inoculated in spring 2018 at tillering stage (Zadoks stage 31)[31]. To inoculate, a 2.5 mL syringe was used to inject an aqueous spore suspension into the leaf bundles, totaling three sequential injections at 7 d intervals. To maintain moisture, mild irrigation followed each inoculation. Susceptible mutants were screened 4–6 wk post inoculation when the control plants of HXH showed full susceptibility to *Pst*. *Pst* tests for other materials were carried out in growth chambers. Inoculations were performed at the seedling stage (Zadoks stage 11–13)[31] by injecting spore suspension into the leaf bundles or by applying a mixture of spores and talc (or *Lycopodium*) powder onto the leaf surface. Inoculated seedlings were kept at 10 °C and 100% relative humidity and in darkness for 24 h, and were then shifted to 16 h light at 15 °C and 8 h darkness at 10 °C. Infection types (ITs) were recorded 15 days post inoculation using a 0–9 scale, where ITs 0–3 were considered resistant, 4–6 intermediate, and 7–9 susceptible[3].

## Phylogenetic analysis

Homologs of *YrNAM* in common wheat and related species were acquired by blasting the *YrNAM* transcripts against the current *Triticeae* genomes in WheatOmics (http://wheatomics.sdau.edu.cn/). Multiple sequence alignments were performed using MAFFT v7.475 L-INS-I (https://mafft.cbrc.jp/alignment/software/) with default parameters. Poorly aligned sequences were removed before constructing a tree. A phylogenetic tree was built by the approximately maximum-likelihood method using FastTree (http://www.microbesonline.org/fasttree/) with default parameters.

## Cellular localization

A GFP fusion construct (PC1221, Ubi::*GFP*:*YrNAM*) was prepared by amplifying the *YrNAM* CDS from P10-46 with the *YrNAM*-F13/R13 PCR primers and the GFP CDS from pMDC43[32] using the GFP-F12/R12 primers. The primers GFP-F12, -R12, and *YrNAM*-R13 each contained a 17-20 bp fragment for homologous recombination. The *GFP* and *YrNAM* were fused and ligated to *BsrG*I linearized PC186[33] using ClonExpress® MultiS One Step Cloning Kit (C113, Vazyme, Nanjing, China). The resultant PC1221 was then transformed into *Agrobacterium* strain EHA105. PC1221 in EHA105 was transiently expressed in tobacco (*Nicotiana benthamiana*) leaves using the infiltration method. After 48 h growth in a controlled environment with a long day photoperiod, the tobacco leaves were photographed using a confocal microscope (Zeiss LSM 880 NLO, Germany). Three views were captured for each target cell, including a bright field, a GFP channel, and a channel for 4′,6-diamidino-2-phenylindole dihydrochloride (DAPI). GFP expression vector pzp211-GFP was used as a control.

## Reporting summary

Further information on research design is available in the Nature Portfolio Reporting Summary linked to this article.

# Data availability

The cDNA and genomic sequence of *YrNAM* have been deposited in NCBI Genbank under the accession numbers OP490604 and OP490605, respectively. DNA resequencing data and error-corrected IsoSeq full-length cDNA reads for resistant WT P10–46, and RNA-Seq data for P10–46 and seven independent mutants have been deposited in the NCBI Sequence Read Archive (SRA) under accession number PRJNA877303. The following public databases were used in this study: IWGSC RefSeq v2.1 [https://wheat-urgi.versailles.inra.fr/Seq-Repository], RNA-Seq expression data of Chinese Spring [https://www.ebi.ac.uk/ena/browser/view/PRJEB5314], PGSB Database [http://pgsb.helmholtz-muenchen.de/plant/index.jsp], *Triticeae* genomes [http://wheatomics.sdau.edu.cn/]. All primers used in this study are listed in Supplementary Data 11. Source data are provided in this paper.

## Code availability

Scripts used for STAM pipeline analysis are available at GitHub.

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

## Acknowledgements

We are grateful to the following people: Ruiming Lin (Chinese Academy of Agricultural Sciences) for providing *Pst* race CYR32 and CYR33, Qingdong Zeng (Northwest A&F University, China) for providing *Pst* race CYR34, Yongtao Zhao (Luohe Academy of Agricultural Sciences, Henan, China) for providing *Pst* mixed spores, Dejun Han (Northwest A&F University) for providing Avocet S and AvSYr10NIL, Honggang Wang (Shandong Agricultural University, China) and Laszlo Purnhauser (Cereal Research Non-profit Ltd., Hungary) for providing wheat germplasm. This work was supported by grants from the Shandong Province Natural Science Foundation of Major Basic Research Program (ZR2021ZD31) and the Youth Innovation Technology Project of Higher School in Shandong Province (2019KJF026) received by J.W., the National Natural Science Foundation of China (31971935) received by F.N., the Agricultural application technology innovation project of Jinan City (CX202113) received by J.W., and the National Transgenic Key Project of the Ministry of Agriculture and Rural Affairs of the People's Republic of China (2016ZX08009003-001-006) received by D.F. and J.W.

## Author contributions

D.F., J.J.W., and F.N. conceptualized the study. J.Z.W., C.Y., and B.L. performed EMS mutagenesis and screened mutants. F.N., Y.Z., and J.L.L. performed STAM analysis. J.J.W. and H.Y. conducted transgenic experiments. X.L, Y.Y., J.J.L., J.Q., and Y.Z. tested Pst resistance of recombinants and transgenic plants. Y.Z., X.M., Y.Y., Q.Y., and Q.Z. investigated *YrNAM* distribution. G.Z. and Y.Z. performed cellular localization analysis. J.J.W., F.N., and D.F. acquired funding. J.J.W., D.F., F.N., and L.E. wrote the paper. All authors approved the final paper.

## Competing interests

The authors declare no competing interests.
