## [Peer Review File · Nature Communications]

Sequencing Trait-Associated Mutations to identify wheat rust-resistance gene YrNAM.Reviewers' Comments:

Reviewer #1:

Remarks to the Author:

The manuscript submitted presented interesting information on the characterization of a new Yr resistance gene encoding NAM and ZnF-BED protein domains.

The identification of NAM and BED protein domains is not new as other cereal disease resistance genes with similar domains have been reported in the literature. I do have a major issue with the identification of the new protein and strongly disagree as being potentially the Yr10 R gene for different reasons: The first one being that Yr10 has been shown a long time ago to be susceptible when challenged with isolate CDL-29 (Beaver and Powelson 1969). This isolate has been used by Line and Chen in numerous subsequent publications. This is the necessary assay to demonstrate unequivocally that the newly characterize gene is in fact, without any doubt, Yr10. Until this work is carried out, the proposed sequence cannot be identified as Yr10. Unless clearly demonstrated, this novel gene represents a new Yr R gene located on the distal portion of wheat chromosome 2BS for which the work was carefully documented. Secondly, non-specific PCR primers were used to probe and follow up Yr10cg (Yuan et al 2012). All PCR primer pairs tested in this publication and cited herein previous publications are targeting the LRR domain, a repetitive region, which always leads to identification of homologs. In addition, we know much better in the field now that the exact R gene sequences are critical for functionality and protecting the challenged plants by pathogen isolates. Divergence by only one nucleotide may be sufficient to change the specificity of any R gene. Based on their previous publications, the authors are well aware of this situation. Sufficient information is provided to reproduce most of the work.

Beaver, R., and Powelson, R. (1969). A new race of stripe rust pathogenic on the wheat variety Moro, CI 13740. *Plant Dis. Rep.* 53, 91–93.

Yuan, C., Jiang, H., Wang, H., Li, K., Tang, H., Li, X., and Fu, D. (2012) Distribution, frequency and variation of stripe rust resistance loci Yr10, Lr34/Yr18 and Yr36 in Chinese wheat cultivars. *J. Genet. Genom.* 39, 587-592.

Reviewer #2:

Remarks to the Author:

Ni and associates report on the cloning of the stripe rust resistance gene Yr10. Remarkably, Yr10 encodes for a protein with NAM and ZnF-BED domains, thus constituting a novel domain architecture in all the nearly 300 plant disease resistance genes cloned to date. Yr10 was identified as a single candidate gene by comparing RNAseq data of independently derived EMS mutants to PacBio Iso-seq data of the wildtype parent. The authors have given the gene cloning approach the acronym STAM for 'sequencing trait associated mutations'. The Yr10 candidate was confirmed by generating stable transgenics in two stripe rust susceptible wheat backgrounds. The manuscript is well written, the figures are clear and didactic, and the statements are generally well backed up by the data provided. Notwithstanding, I have some concerns as detailed below:

1. The abstract does not explain what the basis of STAM is. If this is, as purported, a key selling point of the paper, then the basis of STAM should be made clear in the abstract.
2. Lines 14, 32 and 107: In describing STAM with the terminology "we invented" and "we introduce" the authors do not give credit to earlier reports which have used RNAseq to clone genes in hexaploid wheat, namely (i) Yu et al. 2022 who cloned Sr62 by comparing RNAseq data of independently derived EMS mutants to the RNAseq assembly of a full-length cDNA library of the wildtype non-mutated parent

(see Figure 2, <https://www.nature.com/articles/s41467-022-29132-8>) and (ii) Wang et al. 2022 who cloned Lr9 by comparing RNAseq data of independently derived EMS mutants to the PacBio IsoSeq data of the wildtype non-mutated parent (see Figure 1, https://assets.researchsquare.com/files/rs-1807889/v1_covered.pdf?c=1657038907). The authors should give credit to the prior art in the literature and cite the two above studies appropriately. I would also suggest that the authors document their STAM pipeline (Figure 1a) in e.g. GitHub with precise descriptions of the software versions and parameters used at each step.

3. In lines 24-25 the authors state that the seven previously cloned Yr genes were cloned “mostly by map-based cloning” and then cite the review by Bouvet et al. 2022 to back up this statement. However, Bouvet and colleagues do not review how each of the seven individual Yr genes were cloned. A more appropriate citation would be to Hafeez et al. 2021 (<https://doi.org/10.1016/j.molp.2021.05.014>) who summarize the cloning method for each cloned resistance gene in wheat (see Supplementary Table 2).

4. In line 54 the authors report the exciting novel domain structure of YrNAM, i.e. containing NAM and ZnF-BED domains. Since ZnF-BED domains have been reported in other Poaceae resistance genes, i.e. wheat Yr5, Yr7, barley Rph15 and rice Xo1, the reader is left with the tantalizing question as to how the ZnF-BED domains might or might not be related across all these genes? One gets the feeling that the authors have missed a golden opportunity to conduct a phylogenetic analysis of BED domains as in Figure 3D of Marchal et al (<https://www.nature.com/articles/s41477-018-0236-4>) to establish the evolutionary relationship between R gene associated BED-domains in the Poaceae.

5. It doesn't really make sense to say that YrNAM “evolved from the Sitopsis” section (lines 89 and 104). The connotation of this statement is that YrNAM came from Sitopsis into wheat, i.e. through introgression. However, there is to my knowledge no report of Yr10 having been introgressed from a Sitopsis species into wheat. The alternate (and perhaps the more parsimonious) explanation of the findings of the phylogenetic analysis is that YrNAM and its orthologues in *Ae. sharonensis* and *Ae. longissima* share a common ancestor that predates the speciation of *Ae. sharonensis* and *Ae. longissima*. To support the prior statement of YrNAM “evolving from the Sitopsis” would require additional analysis of the YrNAM sequence and the haplotype in which it resides, and sequence comparison with the syntenic sequences in *Ae. sharonensis* and *Ae. longissima*.

6. Line 380: The authors state that the five Sitopsis species are “related to the B subgenome of common wheat (ref 19)”. Reference 19 refers to “Ma, S. et al. Mol. Plant 14:1965-1968 (2021).” However, it is now commonly accepted that the Sitopsis species *Ae. sharonensis*, *Ae. longissima*, *Ae. bicornis* and *Ae. searsii* are in fact not closely related to the B genome of common wheat – this distinction belongs to only *Ae. speltoides* within the Sitopsis, see e.g. Li, L. et al. Mol Plant 15:488-503 (2022), a paper which the authors also cite in their manuscript.

7. The authors' interpretation of their phylogenetic analysis reported in Supplementary Figure 4 is that YrNAM is absent from the *Ae. bicornis* and *Ae. speltoides* “species” (Lines 382-383). However, an alternate, and possibly more parsimonious scenario is that the species complexes of *Ae. bicornis* and *Ae. speltoides* display presence/absence polymorphisms for YrNAM, similar to what the authors found in common wheat. This alternate hypothesis needs to be stated. Moreover, I would recommend that the authors enrich and strengthen their analysis by including the three other recently reported Sitopsis genome assemblies derived from different accessions to the ones analysed in the YrNAM study. These are *Ae. sharonensis* (Yu et al., 2022; doi: 10.1038/s41467-022-29132-8), and *Ae. speltoides* and *Ae. longissima* (Avni et al., 2022; doi: 10.1111/tpj.15664).

Minor suggestions:

Lines 81 to 84 – the authors describe the physical relationship between YrNAM and markers on chromosome 1B. It would help the reader if the authors provided a supplementary figure to support

this description. Also, the terminology "relatively far from" is vague – please consider providing the actual physical and/or genetic distances.

Figure 2: I would suggest moving the construct used for transformation from Supplementary Figure 2 to the top of Figure 2. Please also consider submitting the PC1213 construct (line 250) to Addgene.

Typos:

Line 81: "Chinese spring" should be "Chinese Spring".

Line 94: "at 3' end" it should be "at the 3' end".

Line 94: "5'end" should be "5' end".

Line 101: Consider "structure" instead of "architecture"?

Lines 196 and 197 and 204: "250bp", "150bp", "153Gb", "18Gb" – should be "250 bp", "150 bp", "153 Gb" and "18 Gb".

Supplementary Figure 4 has a typo in *Ae. sharonensis* 1S-0660.

Reviewer #3:

Remarks to the Author:

The study cloned a yellow rust resistance gene Yr10 in wheat through STAM (Sequencing Trait-Associated Mutations). Specifically, a resistant wildtype was used to build reference transcript sequences via Iso-seq and the candidate transcript was identified through analyzing regular RNA-seq of EMS mutants using reference transcript sequences. The candidate gene was supported by segregation analysis and confirmed with a transgenic experiment. Expression of the resistance allele in susceptible varieties strongly enhanced yellow rust resistance. Evidence provided by this study for the Yr10 cloning is strong. The strategy, STAM, for cloning is straightforward. Below are additional comments:

The major contribution from this study is cloning of Yr10. One of the key experiments for the cloning success is the effort to secure more than 10 EMS mutants that lost resistance. STAM provides an alternative approach to using a reference genome that is available or built through de novo assembly. However, STAM has a risk when the gene is not expressed or expressed at a low level. Given the reduced cost to produce de novo assembly and the assembly quality is high, would STAM be a recommended approach for a new cloning project? Also, can those causal EMS mutations be identified if RNA-seq reads from mutants are mapped to a reference genome (e.g., the Chinese Spring reference genome) or genomic assembled contigs produced in this study? If not, reasoning to use STAM can be strengthened.

In general, the manuscript could be improved by precisely describing some genetic terms. For example, the resistance allele should be a dominant allele, which is not described; WT P10-46 can be clarified as "a resistant WT"; "All plants with a homozygous C1142T genotype were susceptible" could replace L69-70.

L18: Might be better to replace cisgenics with a more general term

L49-50: It is not clear what data were used for de novo genome assembly. Also, how did those contigs end up with non-repetitive DNA sequences?

In Supplemental Fig. 2, does 5'-UTR and 3'-UTR represent upstream and downstream sequences of the gene, respectively? If yes, UTRs need to be modified.

Thank you very much for allowing us to revise the manuscript. We have highlighted the revised content in the manuscript with track changes. In the manuscript, all changes were in response to reviewer comments, except that we revised & added sentences at the end of the abstract because our original abstract was shorter than necessary for *Nature Communications*. The revised and additional sentences at the end of the abstract state, “Using STAM with seven mutant wheat lines that have a non-functional wheat stripe rust resistance gene *Yr10*, we show that *Yr10* encodes for a protein (named *YrNAM*) with a NAM and a ZnF-BED domain; based on 12 independent mutants, both domains are necessary for function. Using a vector that encodes for *YrNAM* with its endogenous promoter, we used stable transgenesis to experimentally demonstrate that *YrNAM* confers resistance to *Puccinia striiformis* f. sp. *tritici* CYR32 and CYR33. Segregation analysis indicates that *YrNAM* is *Yr10*.”

We here provide point-by-point responses to the reviewers. References cited here are listed at the end of this file.

REVIEWER COMMENTS IN GREEN
AUTHOR RESPONSE IN BLUE

I. Reviewer #1 (Remarks to the Author):

Overall impression:

The manuscript submitted presented interesting information on the characterization of a new Yr resistance gene encoding NAM and ZnF-BED protein domains. The identification of NAM and BED protein domains is not new as other cereal disease resistance genes with similar domains have been reported in the literature.

Response: Certainly, either NAM or ZnF-BED were separately reported in plant disease resistance (R) proteins, e.g. NAM in *ANAC042/JUB 1*(1), *NAC4* (1), *TaNAC21/22* (1) and *TuNAC69* (2), and ZnF-BED in *Rhp15* (3), *Xo1* (4, 5) and *Yr7* (6). However, all 47 known R proteins in wheat lack a NAM domain (7), and some ZnF-BED-containing R proteins also have a canonical NLR domain. We now make this point explicitly in line 73-77. To our knowledge, we here reveal a novel R gene architecture with both NAM and ZnF-BED domains. If this novel architecture of a R gene was previously reported, we will cite any such work.

Major comments:

I-1. I do have a major issue with the identification of the new protein and strongly disagree as being potentially the Yr10 R gene for different reasons: The first one being that Yr10 has been shown a long time ago to be susceptible when challenged with isolate CDL-29 (Beaver and Powelson 1969). This isolate has been used by Line and Chen in numerous subsequent publications. This is the necessary assay to demonstrate unequivocally that the newly characterize gene is in fact, without any doubt, Yr10. Until this work is carried out, the proposed sequence cannot be identified as Yr10. Unless clearly demonstrated, this novel gene represents a new Yr R gene located on the distal portion of wheat chromosome 2BS for which the work was carefully documented.

Response: In hindsight, perhaps our description of previous work on *Yr10CG* was insufficient. We have now expanded this paragraph with more on the erroneous publication, and the identifying features of *Yr10*. The augmented text now reads, “*Yr10*, from a Turkish wheat ‘PI 178383’, confers all-stage resistance to *Pst*, is deployed in wheat cultivar ‘Moro’³, and is present in AvSYr10NIL⁴. Wheat with *Yr10* are resistant to multiple *Pst* strains including CYR29, CYR31, CYR32, and CYR33, but susceptible to CYR34⁴⁻⁶. Based on preliminary evidence, Laroche and colleagues proposed that a CC-NBS-LRR gene (Genbank AF149112) was a candidate for *Yr10*⁷⁻⁸. However, our haplotype and linkage analyses for AF149112, which we called *Yr10CG*, demonstrated that *Yr10CG* is not *Yr10*⁹⁻¹⁰; for example, *Yr10CG* is 1.2 cM away from *Yr10*, and wheat lines that express AF149112 are susceptible to CYR29.”

As additional information to the reviewers, *Yr10* is on 1BS, not on 2BS as indicated above. We reviewed the paper by Beaver and Powelson (8), and there is no mention of CDL-29 in that paper. Their only virulent race on Moro was W-57; perhaps W-57 and CDL-29 are the same. Because the reviewer states, “This isolate has been used by Line and Chen in numerous subsequent publications”, we wrote to Dr. Xianming Chen (the reviewer’s cited author), who is the most famous wheat stripe rust pathologist in the US and who maintains the best wheat stripe rust collection of US; we asked Dr. Chen about CDL-29 and W-57. According to Dr. Chen, the old race CDL-29 is equivalent to PSTv-23 (synonym, PST-29). CDL-29 is a fairly old race. We are trying to recover it from an over one-decade storage that were acquired from USA. Back on then, it was fairly easy

to exchange races even without written permission. However, today, trying to import these races from the U.S. is not a reasonable strategy for us because the paperwork takes a long time and there's no guarantee that these strains would ever be approved for import. In addition, PSTv-23 (=CDL-29) should be avirulent to *Yr6* (9), a seedling resistance gene that is in wheat cultivar 'Fielder' (9, 10); PSTv-23 is also avirulent to *Yr9* (9), a seedling resistance gene that occurs in particular genotype of wheat 'CB037' (11). In our study, the genetic complementation was done in the Fielder and particular CB037 genetic backgrounds, which potentially precludes the use of PSTv-23 (=CDL-29) to corroborate *Yr10*.

However, Dr. Chen suggested that we use Chinese races, both *Yr10*-avirulent and *Yr10*-virulent races, to make sure that we are using the correct lines for *Yr10*. We now have completed additional experiments and believe that we have fully done this. We have done one trial with *Yr10*-avirulent race CYR33 and two independent trials with *Yr10*-virulent race CYR34. We now state on lines 94-99, "In T₂ generation, PC1213 transgenic plants showed resistance to CYR33, an avirulent race for *Yr10* as documented in Moro, P8-13, and P10-46 (Fig. 3a). However, CYR34 is virulent to *Yr10* because it caused abundant sporulation in Moro, P8-13, and P10-46 (Fig. 3b). Unexpectedly, PC1213 transgenic T₂ plants showed resistance (ITs=2) to CYR34 (Fig. 3b), which were likely caused by pyramiding *Yr10* and *Yr6* in Fielder¹⁷ and *Yr9* in CB037¹⁸".

I-2. Secondly, non-specific PCR primers were used to probe and follow up *Yr10cg* (Yuan et al 2012). All PCR primer pairs tested in this publication and cited herein previous publications are targeting the LRR domain, a repetitive region, which always leads to identification of homologs. In addition, we know much better in the field now that the exact R gene sequences are critical for functionality and protecting the challenged plants by pathogen isolates. Divergence by only one nucleotide may be sufficient to change the specificity of any R gene. Based on their previous publications, the authors are well aware of this situation.

Response: We completely agree that the sequence of a R gene governs its specificity

and functionality, and that, indeed, one needs extreme caution with primers that are in a repetitive region. However, the reviewer is incorrect about (presumably) our two diagnostic markers *Yr10_{CG}E2a* and *Yr10_{CG}E2b* (12), which were designed to detect the presence of AF149112 (Genbank) (13). To clarify, *Yr10_{CG}E2a* and *Yr10_{CG}E2b* (12) are specific for a CC-NBS-LRR gene associated with AF149112, which we named *Yr10_{CG}* and is not *Yr10* (12, 14). Below, we illustrate why *Yr10_{CG}E2a* and *Yr10_{CG}E2b* are specific for AF149112.

First, AF149112 encodes a CC-NBS-LRR protein with 824 aa, of which LRR spans from 486 to 757 aa (Fig. 1 below).

Fig.1 Motif and marker region in the AF149112-corresponding protein

Yr10_{CG}E2a and *Yr10_{CG}E2b* match 676-799 aa and 698-769 aa, respectively. Thus, the reverse primers of both markers fall outside of the LRR domain, which was purposely designed for specificity for the AF149112 sequence. Although one PCR marker is often sufficient to correctly genotype a gene, because we were dealing with a CC-NBS-LRR sequence with multiple homologs in the wheat genome, we designed two markers and used both to avoid any confusion. In practice, *Yr10_{CG}E2a* and *Yr10_{CG}E2b* are very specific, and we only conclude that AF149112 is present when both markers were positive. In other cases, depending on the goals, the LRR domain alone was used to develop PCR makers, e.g. for *Pm3a* and *Pm3f* (15).

Second, PCR primers were designed by using AF149112-specific bases versus its close homologues (Fig. S1 in (12)). The 3' end of each primer normally contain two or more continuous bases specific to AF149112. In this case, the reverse primer of *Yr10_{CG}E2a* contains a 10bp insertion specific to AF149112 (Fig. S1 in (12)). Furthermore, we searched *Yr10_{CG}E2a* (371 bp) in currently accessible wheat genomes, which contain far more sequence data than available in 2012 (12). As shown in Fig. 2A below, there are

many hits for *Yr10_{CGE2a}* in WheatOmics (<http://wheatomics.sdau.edu.cn/>), however the reverse primer (from 351 to 371 bp) of *Yr10_{CGE2a}* is highly divergent among all close hits, except for two hits from ‘Jagger’ (chr. Un) and ‘Norin 61’ (chr. 1B) (Fig. 2A). We then searched the full-length sequence of AF149112 in 18 wheat genotypes (Fig. 2B); the best hits from Norin 61 (Fig. 2B) and Jagger (Fig. 2C) share more than 99.8% sequence identity versus AF149112, but the top hits from the other 16 genotypes share less than 94% sequence identity versus AF149112. Certainly, the AF149112 sequence occurs in Jagger and Norin 61, but not in other genotypes. Therefore, *Yr10_{CGE2a}* is specific for the AF149112 fragment in wheat. From this study, we already knew that the *YrNAM* gene, which represents *Yr10*, is absent in all 18 wheat genotypes, although there are some >1kb fragments sharing a sequence identity up to 81% versus *YrNAM*.

The AF149112-corresponding sequence in Jagger is partially incomplete in the 5’- and 3’-ends, likely caused by poor assembly. The AF149112-corresponding sequence is well conserved in Norin 61, which has four SNPs when compared to ‘Moro’. Of the four SNPs between Norin 61 and Moro, three SNPs occur in the intron between splicing sites, and the only SNP in the exon is synonymous for protein coding. Likely, if AF149112 did represent *Yr10* (it doesn’t), Norin 61 should have *Yr10* too. Norin 61 was registered in 1944 (16), with a pedigree of Fukuoka-komugi18/Shinchunaga (17). However, the *Yr10* donor germplasm PI178383 was first collected from Turkey in 1948 (18); the first *Yr10* cultivar Moro was registered in 1966, with a pedigree of PI178383/Omar (19). Therefore, it is unlikely that there is a *Yr10* gene in Norin 61 that is derived from PI178383 or Moro.

Third, The PCR specificity was tested on isogenic lines of ‘Avocet S’, ‘Avocet R’ and ‘Avocet S+Yr10’. Both *Yr10_{CGE2a}* and *Yr10_{CGE2b}* only worked in Avocet S+Yr10, but not in Avocet S and Avocet R (Fig. S2 (12)). The specificity was further confirmed by sequencing the PCR products from Avocet S+Yr10, Moro, Nanda2419 and Jiangdongmen (12).

All together, we are extremely careful about the specificity of our primers, and are confident that *Yr10_{CGE2a}* and *Yr10_{CGE2b}* are indeed specific for the AF149112 fragment.

Fig.2A Blastn hits of the 371 bp fragment of *Yr10cgE2a*

Descriptions

Sequences producing significant alignments:

Description	Max score	Total score	Query cover	E value	Ident	Accession
Norin61 chromosome chr1B	6529.5	17911.8	109%	0	100%	chr1B_Norin61
chromosome chr1B	4912.8	25183.9	112%	0	93%	chr1B_Renan
Mace chromosome chr1B	4912.8	25712.2	112%	0	93%	chr1B_Mace
LongReach Lancer chromosome chr1B	4912.8	13337.5	111%	0	93%	chr1B_LongReach_Lancer
Kariega chr1B chromosome	4912.8	34020.4	116%	0	93%	chr1B_Kariega
Wild emmer chromosome chr1B	4905.6	20764.7	109%	0	90%	chr1B_Wild_emmer
Jagger chromosome chr1B	2252.8	11527.0	107%	0	93%	chr1B_Jagger
Zang1817 chromosome chr1B	1994.0	6247.6	75%	0	87%	chr1B_Zang1817
SY Mattis chromosome chr1B	1994.0	7883.2	83%	0	93%	chr1B_SY_Mattis
Julius chromosome chr1B	1994.0	7907.6	83%	0	93%	chr1B_Julius
Durum Wheat chromosome chr1B	1994.0	7814.7	81%	0	93%	chr1B_Durum_Wheat
Chinese Spring1.0 chromosome chr1B	1994.0	6112.3	75%	0	87%	chr1B_Chinese_Spring1.0
CDC Stanley chromosome chr1B	1994.0	7965.3	80%	0	93%	chr1B_CDC_Stanley
Attraktion chr1B chromosome	1994.0	8553.2	89%	0	93%	chr1B_Attraktion
Fielder chromosome chr1B	1100.4	6083.5	84%	0	93%	chr1B_Fielder
CDC Landmark chromosome chr1B	1100.4	5053.8	77%	0	93%	chr1B_CDC_Landmark
KN9204 chr1B chromosome	151.9	3296.4	51%	1.626e-32	77%	chr1B_KN9204
ArinaLrFor chromosome chr1B	144.7	3208.0	51%	2.413e-30	75%	chr1B_ArinaLrFor

Fig.2B Chromosome 1B hits of the AF149112 fragment (3630 bp)

Fig.2C Blastn hits of the AF149112 fragment (3630 bp) in 'Jagger'

I-3. Sufficient information is provided to reproduce most of the work.

Response: Thank you!

II. Reviewer #2 (Remarks to the Author):

Overall impression:

Ni and associates report on the cloning of the stripe rust resistance gene Yr10. Remarkably, Yr10 encodes for a protein with NAM and ZnF-BED domains, thus constituting a novel domain architecture in all the nearly 300 plant disease resistance genes cloned to date. Yr10 was identified as a single candidate gene by comparing

RNAseq data of independently derived EMS mutants to PacBio Iso-seq data of the wildtype parent. The authors have given the gene cloning approach the acronym STAM for ‘sequencing trait associated mutations’. The Yr10 candidate was confirmed by generating stable transgenics in two stripe rust susceptible wheat backgrounds. The manuscript is well written, the figures are clear and didactic, and the statements are generally well backed up by the data provided. Notwithstanding, I have some concerns as detailed below:

Response: Thanks! We appreciate reading “Remarkably, *Yr10* encodes for a protein with NAM and ZnF-BED domains, thus constituting a novel domain architecture in all the nearly 300 plant disease resistance genes cloned to date.”

Major comments:

II.1 The abstract does not explain what the basis of STAM is. If this is, as purported, a key selling point of the paper, then the basis of STAM should be made clear in the abstract.

Response: Thank you for this comment. We added a new third sentence to the abstract, “STAM identifies gene candidates by mapping RNA-Seq data from multiple, independent, trait-specific mutants onto a full-length reference transcriptome of the wild-type; transcripts with mutations in multiple mutants are investigated further.”

II.2 Lines 14, 32 and 107: In describing STAM with the terminology “we invented” and “we introduce” the authors do not give credit to earlier reports which have used RNAseq to clone genes in hexaploid wheat, namely (i) Yu et al. 2022 who cloned Sr62 by comparing RNAseq data of independently derived EMS mutants to the RNAseq assembly of a full-length cDNA library of the wildtype non-mutated parent (see Figure 2, <https://www.nature.com/articles/s41467-022-29132-8>) and (ii) Wang et al. 2022 who cloned Lr9 by comparing RNAseq data of independently derived EMS mutants to the PacBio IsoSeq data of the wildtype non-mutated parent (see Figure 1, https://assets.researchsquare.com/files/rs-1807889/v1_covered.pdf?c=1657038907). The authors should give credit to the prior art in the literature and cite the two above studies appropriately. I would also suggest that the authors document their STAM pipeline (Figure 1a) in e.g. GitHub with precise descriptions of the software versions and parameters used at each step.

Response: Thanks for the constructive comments. We apologize for omitting them. We now cite both in Line 50: “Recently, similar approaches have been used to clone wheat rust resistance genes *Lr9*¹¹ and *Sr62*¹²”. As requested, we added precise descriptions on GitHub (<https://github.com/Feiny/STAM>), and added a note in Data availability. And we have changed “We invented” to “We developed” and “we introduce” to “we used”.

II.3 In lines 24-25 the authors state that the seven previously cloned Yr genes were cloned “mostly by map-based cloning” and then cite the review by Bouvet et al. 2022 to back up this statement. However, Bouvet and colleagues do not review how each of the seven individual Yr genes were cloned. A more appropriate citation would be to Hafeez et al. 2021 (<https://doi.org/10.1016/j.molp.2021.05.014>) who summarize the cloning method for each cloned resistance gene in wheat (see Supplementary Table 2).

Response: Excellent! We changed the citation to Hafeez et al. 2021.

II.4 In line 54 the authors report the exciting novel domain structure of YrNAM, i.e. containing NAM and ZnF-BED domains. Since ZnF-BED domains have been reported in other Poaceae resistance genes, i.e. wheat Yr5, Yr7, barley Rph15 and rice Xo1, the reader is left with the tantalizing question as to how the ZnF-BED domains might or might not be related across all these genes? One gets the feeling that the authors have missed a golden opportunity to conduct a phylogenetic analysis of BED domains as in Figure 3D of Marchal et al (<https://www.nature.com/articles/s41477-018-0236-4>) to establish the evolutionary relationship between R gene associated BED-domains in the Poaceae.

Response: Thanks for your valuable comment. We phylogenetically analyzed R gene associated BED-domains in *Poaceae* and now present the data in Supplemental Figure 8 and Supplemental Figure 9. We added the following sentences to Line 111-117:

The Znf-BED domains were highly conserved among these *YrNAM* orthologs/homologs, especially among genes annotated in chromosomes 1A, 1B and 1S (Supplemental Fig. 8). In contrast, there was 37-44% identity between the YrNAM and the ZnF-BED domains of the six characterized NLR-BED genes for disease resistance in the *Poaceae*¹³⁻¹⁶. The phylogenetic tree analysis showed the ZnF-BED domain of YrNAM was separated from those ZnF-BED domains of NLR proteins in *Poaceae* (Supplemental Fig. 9).

II.5 It doesn't really make sense to say that YrNAM "evolved from the Sitopsis" section (lines 89 and 104). The connotation of this statement is that YrNAM came from Sitopsis into wheat, i.e. through introgression. However, there is to my knowledge no report of Yr10 having been introgressed from a Sitopsis species into wheat. The alternate (and perhaps the more parsimonious) explanation of the findings of the phylogenetic analysis is that YrNAM and its orthologues in *Ae. sharonensis* and *Ae. longissima* share a common ancestor that predates the speciation of *Ae. sharonensis* and *Ae. longissima*. To support the prior statement of YrNAM "evolving from the Sitopsis" would require additional analysis of the YrNAM sequence and the haplotype in which it resides, and sequence comparison with the syntenic sequences in *Ae. sharonensis* and *Ae. longissima*.

Response: Another very constructive comment! We have modified Line 106-111 as "The phylogenetic analysis (Supplemental Fig. 5, 6, Supplemental Data 6) and molecular marker investigation (Supplemental Fig. 7, Supplemental Data 7) revealed orthologs and/or close homologs of *YrNAM* in *Aegilops longissima* and *Ae. sharonensis*, suggesting that the origin of *YrNAM* predates the speciation of wheat B- and D-subgenome progenitors, probably 7.3 million years ago¹⁹". We also revised Line 132-133 as "In addition, we showed that *YrNAM* likely evolved before the speciation of wheat B- and D-subgenome progenitors, is only present in a few wheat cultivars..."

II.6 Line 380: The authors state that the five Sitopsis species are "related to the B subgenome of common wheat (ref 19)". Reference 19 refers to "Ma, S. et al. Mol. Plant 14:1965-1968 (2021)." However, it is now commonly accepted that the Sitopsis species *Ae. sharonensis*, *Ae. longissimi*, *Ae. bicornis* and *Ae. searsii* are in fact not closely related to the B genome of common wheat – this distinction belongs to only *Ae. speltoides* within the Sitopsis, see e.g. Li, L. et al. Mol Plant 15:488-503 (2022), a paper which the authors also cite in their manuscript.

Response: Thanks. We deleted this statement.

II.7 The authors' interpretation of their phylogenetic analysis reported in Supplementary Figure 4 is that YrNAM is absent from the *Ae. bicornis* and *Ae. speltoides* "species" (Lines 382-383). However, an alternate, and possibly more

parsimonious scenario is that the species complexes of *Ae. bicornis* and *Ae. speltoides* display presence/absence polymorphisms for YrNAM, similar to what the authors found in common wheat. This alternate hypothesis needs to be stated. Moreover, I would recommend that the authors enrich and strengthen their analysis by including the three other recently reported *Sitopsis* genome assemblies derived from different accessions to the ones analysed in the YrNAM study. These are *Ae. sharonensis* (Yu et al., 2022; doi: 10.1038/s41467-022-29132-8), and *Ae. speltoides* and *Ae. longissima* (Avni et al., 2022; doi: 10.1111/tpj.15664).

Response: Great! We revised as “In all five *Sitopsis* species, *YrNAM* homologs are present in *Ae. longissima*, *Ae. sharonensis*, and *Ae. searsii* (cv. TE01) (Supplemental Fig. 5), but undetectable in *Ae. bicornis* accession TB01¹⁹ and *Ae. speltoides* accessions TS01¹⁹ and AEG-9674-1²⁹. Given whole species complexes, it is possible that *Ae. bicornis* and/or *Ae. Speltoides* associate with a presence/absence pattern for *YrNAM* homologs.”

As for the new genome resources, we previously included them in the Supplemental Data 6. Although an annotated gene was not retrieved in *Ae. longissimi* AEG-6782-2, there were multiple hits in blastn search against the genomic sequences. We added notes for the references and the blastn results against genomic DNAs in Supplemental Data 6.

II. 9 Minor suggestions:

Lines 81 to 84 – the authors describe the physical relationship between YrNAM and markers on chromosome 1B. It would help the reader if the authors provided a supplementary figure to support this description. Also, the terminology “relatively far from” is vague – please consider providing the actual physical and/or genetic distances.

Response: The supplemental Figure 3 has been added. We replaced “relatively far from” with “around 1.0 Mb”.

Figure 2: I would suggest moving the construct used for transformation from Supplementary Figure 2 to the top of Figure 2. Please also consider submitting the PC1213 construct (line 250) to Addgene.

Response: Construct PC1213 was moved to Figure 2. We are providing the construct to Addgene. In addition, upon request, we will provide the construct for research purposes at no charge. We added this note in the Data availability.

Typos:

Line 81: “Chinese spring” should be “Chinese Spring” .

Response: Corrected.

Line 94: “at 3’ end” it should be “at the 3’ end” .

Response: Corrected.

Line 94: “5’ end” should be “5’ end” .

Response: Corrected.

Line 101: Consider “structure” instead of “architecture” ?

Response: Done.

Lines 196 and 197 and 204: “250bp”, “150bp”, “153Gb”, “18Gb” – should be “250 bp”, “150 bp”, “153 Gb” and “18 Gb” .

Response: All corrected.

Supplementary Figure 4 has a typo in *Ae. sharonensis* IS-0660.

Response: Corrected.

III. Reviewer #3 (Remarks to the Author):

Overall impression:

The study cloned a yellow rust resistance gene Yr10 in wheat through STAM (Sequencing Trait-Associated Mutations). Specifically, a resistant wildtype was used to build reference transcript sequences via Iso-seq and the candidate transcript was identified through analyzing regular RNA-seq of EMS mutants using reference transcript sequences. The candidate gene was supported by segregation analysis and confirmed with a transgenic experiment. Expression of the resistance allele in susceptible varieties strongly enhanced yellow rust resistance. Evidence provided by this study for the Yr10 cloning is strong. The strategy, STAM, for cloning is straightforward.

Response: Thank you! We are particularly grateful for your comments, “Evidence provided by this study for the *Yr10* cloning is strong. The strategy, STAM, for cloning is straightforward.”

Major comments:

III.1 The major contribution from this study is cloning of *Yr10*. One of the key experiments for the cloning success is the effort to secure more than 10 EMS mutants that lost resistance. STAM provides an alternative approach to using a reference genome that is available or built through de novo assembly. However, STAM has a risk when the gene is not expressed or expressed at a low level. Given the reduced cost to produce de novo assembly and the assembly quality is high, would STAM be a recommended approach for a new cloning project? Also, can those causal EMS mutations be identified if RNA-seq reads from mutants are mapped to a reference genome (e.g., the Chinese Spring reference genome) or genomic assembled contigs produced in this study? If not, reasoning to use STAM can be strengthened.

Response: Thanks! The reviewer raises interesting and important questions about STAM technology. At the end of the last paragraph, we have added the following text, “For cloning *Yr10*, STAM was a low-cost option; *YrNAM* is absent in assembled wheat genomes; *de novo* genomic assemblies of complex, polyploid genomes such as wheat are still expensive; and, in general, independent, EMS mutants in wheat are relatively easy to obtain, particularly when the trait is encoded by a single gene. STAM may not be the best option for identifying genes for polygenic traits, and is unsuitable for identifying genes with mutations in regulatory parts of a gene. We note that we identified *Yr10* with STAM even though *Yr10* apparently, had low expression, which might be due to highly localized expression, with only 8 circular consensus sequencing (CCS) reads in 20 Gb IsoSeq (~3.3 million CCS reads) from the wild type and on average 1.56-4.21 transcripts per million (TPM) in RNA-Seq of the WT at different time points after *Pst* inoculation”.

III.2 In general, the manuscript could be improved by precisely describing some genetic terms. For example, the resistance allele should be a dominant allele, which is not described; WT P10-46 can be clarified as “a resistant WT” ; “All plants with a homozygous C1142T genotype were susceptible” could replace L69-70.

Response: Thanks! We modified the Line 128, “*Yr10* has a previously unidentified architecture” was replaced by “The dominant resistance allele has a previously unidentified structure”. WT P10-46 was clarified as “a resistant WT” in Line 52, 82, 105, 160, 278, and 391. Line 83-84 has been replaced by “All plants with a homozygous C1142T genotype were susceptible”.

III.3 L18: Might be better to replace cisgenics with a more general term.

Response: Replaced.

III.4 L49-50: It is not clear what data were used for de novo genome assembly. Also, how did those contigs end up with non-repetitive DNA sequences?

Response: The *de novo* genome assembly was described in Line 286-292. Basically, Illumina NovaSeq (2×150 bp reads) sequencing data were filtered to remove repetitive DNAs by mapping to the Poaceae repeat element database mipsREdat 9.3p. The resultant 44.2 Gb non-repetitive DNA sequences were used for de novo assembly using SPADSV3.15. The result was described in Line 63: “*de novo* assembled ~1.1M contigs of non-repetitive DNA with 1.8 Gb in total”.

III.5 In Supplemental Fig. 2, does 5’ -UTR and 3’ -UTR represent upstream and downstream sequences of the gene, respectively? If yes, UTRs need to be modified.

Response: Corrected.

References

1. Yuan, X., et al. NAC transcription factors in plant immunity. *Phytopathol Res.* **1**, 3 (2019).
2. Xu Y., et al. A NAC transcription factor *TuNAC69* contributes to ANK-NLR-WRKY NLR-mediated stripe rust resistance in the diploid wheat *Triticum urartu*. *Int. J. Mol. Sci.* **23**, 564 (2022).
3. Chen C., et al. BED domain-containing NLR from wild barley confers resistance to leaf rust. *Plant Biotech J.* **19**, 1206-1215 (2021).
4. Read A., et al. Cloning of the rice *Xo1* resistance gene and interaction of the Xo1 protein with the defense-suppressing *Xanthomonas* effector *Tal2h*. *Mol. Plant Microbe Interact.* **33**, 1189-1195 (2020).

5. Read A., et al. Genome assembly and characterization of a complex zfBED-NLR gene-containing disease resistance locus in Carolina Gold Select rice with Nanopore sequencing. *PLoS Genet.* **16**, e1008571 (2020).
6. Marchal, C., et al. BED-domain-containing immune receptors confer diverse resistance spectra to yellow rust. *Nat. Plants* **4**, 662-668 (2018).
7. Hafeez A.N., et al. Creation and judicious application of a wheat resistance gene atlas. *Mol. Plant* **14**, 1053-1070 (2021).
8. Beaver R.G. & Powelson R.L. A new race of stripe rust pathogenic on the wheat variety Moro, CI 13740. *Plant Disease Reporter* **53**, 91-93 (1969).
9. Wan A.M. & Chen X.M. Virulence characterization of *Puccinia striiformis* f. sp. *tritici* using a new set of *Yr* single-gene line differentials in the United States in 2010. *Plant Dis.* **98**, 1534-1542 (2014).
10. Line, R. F., & Qayoum, A. Virulence, aggressiveness, evolution and distribution of races of *Puccinia striiformis* (the cause of stripe rust of wheat) in North America, 1968-87. *US Dep. Agric Tech. Bull.* **1788**, 1-44 (1992).
11. Liu, H. et al. Identification of three wheat near isogenic lines originated from CB037 on tissue culture and transformation capacities. *Plant Cell Tiss. Organ Cult.* **152**, 67-79 (2023).
12. Yuan C., et al. Distribution, frequency and variation of stripe rust resistance loci *Yr10*, *Lr34/Yr18* and *Yr36* in Chinese wheat cultivars. *J. Genet. Genomics* **39**, 587-592 (2012).
13. Liu W., et al. The stripe rust resistance gene *Yr10* encodes an evolutionary-conserved and unique CC-NBS-LRR sequence in wheat. *Mol. Plant* **7**, 1740-1755 (2014).
14. Yuan C., et al. Remapping of the stripe rust resistance gene *Yr10* in common wheat. *Theor. Appl. Genet.* **131**, 1253-1262 (2018).
15. Tommasini L., et al. Development of functional markers specific for seven Pm3 resistance alleles and their validation in the bread wheat gene pool. *Theor. Appl. Genet.* **114**, 165-175 (2006).
16. Genetic studies of photoperiod response genes and their effect on heading time in Japanese wheat cultivars (2015 https://www.naro.go.jp/publicity_report/publication/archive/laboratory/nics/report/057305.html).
17. Ban T. & Suenaga K. Genetic analysis of resistance to *Fusarium* head blight caused by *Fusarium graminearum* in Chinese wheat cultivar Sumai 3 and the Japanese cultivar

Saikai 165. *Euphytica* **113**, 87-99 (2000).

18. Harlan J.R. Disease as a factor in plant evolution. *Annu. Rev. Phytopathol.* **14**: 31-51 (1976).
19. Rohde R. Registration of Moro wheat. *Crop Sci.* **6**, 502 (1966).

Reviewers' Comments:

Reviewer #2:

Remarks to the Author:

The authors have made great improvements to the manuscript and do not appear to have cut any corners to address the concerns I had with their first version. I only have a few minor comments:

1. Following on from the new ZnF-BED phylogenetic analysis the authors write: "The phylogenetic tree analysis showed the ZnF-BED domain of YrNAM was separated from those ZnF-BED domains of NLR proteins in Poaceae (Supplemental Fig. 9)"

To me this analysis provides clear evidence that independent ZnF-BED domain insertions occurred in NLRs, e.g. Rph15/Yr5/Yr7 vs YrNAM. This is an intriguing observation and might perhaps be worth mentioning.

2. The authors have introduced the new sentence: "Given whole species complexes, it is possible that *Ae. bicornis* and/or *Ae. Speltoides* associate with a presence/absence pattern for YrNAM homologs." This sentence is poorly written and could be improved for clarity.

On the other hand, I also read the other reviewer's comments and the rebuttals by the authors. In the revised version, the authors state in response to Reviewer 1 that "Unexpectedly, PC1213 transgenic T2 plants showed resistance (ITs=2) to CYR34 (Fig. 3b), which were likely caused by pyramiding Yr10 and Yr6 in Fielder (ref 17) and Yr9 in CB037 (ref 18)".

If this 'masking' assumption is correct, then CYR34 should be avirulent on cv Fielder, however, strangely, CYR34 is virulent on cv Fielder (see Fig. 3B). This indicates that the transgenics are no longer race specific, possibly suggesting that they express a non-specific ectopic resistance, e.g. due to over-expression. This is actually quite an important point viz-a-viz the identity of the cloned gene and its relationship with Yr10. In conclusion, I am unable to reconcile this observation with YrNAM being Yr10. I feel this requires further consideration and look forward to see what Reviewer 1 has to say about this.

Reviewer #3:

Remarks to the Author:

My previous concerns/comments have been well addressed. Here are two minor comments for the revision:

L95: "However, CYR34 is virulent to Yr10 because it caused abundant sporulation in Moro, P8-13, and P10-46 (Fig. 3b). Unexpectedly, PC1213 transgenic T2 plants showed resistance (ITs=2) to CYR34 (Fig. 3b), which were likely caused by pyramiding Yr10 and Yr6 in Fielder17 and Yr9 in CB03718." In the revision, the authors hypothesized that Fielder carrying Yr6 caused the resistance to CYR34. Since Fielder was used as the control, why not present data of the controls to support the hypothesis?

L151: What is localized expression in "which might be due to highly localized expression"?

Reviewer #4:

Remarks to the Author:

There remain two questions should be addressed to identify the candidate gene as Yr10.

Firstly, according to the gene-for-gene theory, the race-specific resistance genes containing Yr10 will

initiate plant disease resistance response only when they recognize the corresponding avirulent genes. Results in this study showed that YrNAM transgenic wheat lines were high resistant to all the tested Pst races comprise of CYR29, CYR31, CYR32, CYR33 and CYR34 as well as a mixture of spores from naturally infected wheat fields, thus, which could not be defined as Yr10. Secondly, Yr10 was identified and named a long time ago using different Pst races, and the races used in this study are different from those at that time. Therefore, to define the real Yr10 also needs to choose the same Pst races to verify.

We thank the editor and all reviewers. We believe that we have fully complied with the editor's option 2. The last sentence from the abstract "Segregation analysis indicates that *YrNAM* is *Yr10*" has been deleted. In the text about our current results, *Yr10* has been replaced with *YrNAM*. In text about our previously published results, we now generally refer to *Yr10* as the *Yr10* region.

We have added a new paragraph in the Discussion:

"The *Yr10* region currently has two fully identified genes for stripe rust resistance in cv. Moro and AvSYr10NIL: *Yr10CG* and *YrNAM*, which are 1.2 cM and 1.0 Mb apart (Supplementary Fig. 3). Our previous¹⁰ and current study indicates that *YrNAM* might be *Yr10*. However, whether the *Yr10* activity is due to *Yr10CG*, *YrNAM* or both remains to be tested. Regardless, the Wheat Gene Atlas² has called *Yr10CG* a synonym for *Yr10*, and research in our study has been done with Chinese strains of *Pst*¹⁰, and not with the historic North American strains that were used to characterize *Yr10*³. Consequently, we here name *YrNAM* as providing resistance to *Pst* strains CYR29, CYR31, CYR32 and CYR33. We also note that *YrNAM* has some of the breeding value that was previously attributed to *Yr10*²¹".

We here provide point-by-point responses to the reviewers.

REVIEWER COMMENTS IN GREEN

AUTHOR RESPONSE IN BLUE

I. Reviewer #2

The authors have made great improvements to the manuscript and do not appear to have cut any corners to address the concerns I had with their first version. I only have a few minor comments:

1. Following on from the new ZnF-BED phylogenetic analysis the authors write: "The phylogenetic tree analysis showed the ZnF-BED domain of *YrNAM* was separated from those ZnF-BED domains of NLR proteins in Poaceae (Supplemental Fig. 9)".

To me this analysis provides clear evidence that independent ZnF-BED domain insertions occurred in NLRs, e.g. Rph15/Yr5/Yr7 vs YrNAM. This is an intriguing observation and might perhaps be worth mentioning.

Response: Thank you for this suggestion. We now state, “The phylogenetic tree analysis showed the ZnF-BED domain of YrNAM is distinct from the ZnF-BED domains of NLR proteins in *Poaceae* (Supplemental Fig. 9). **Thus, the ZnF-BED domain insertion in YrNAM is independent from the ones in NLRs Rph15 and Yr5/Yr7.**

2. The authors have introduced the new sentence: "Given whole species complexes, it is possible that *Ae. bicornis* and/or *Ae. Speltoides* associate with a presence/absence pattern for YrNAM homologs." This sentence is poorly written and could be improved for clarity.

Response: We apologize. Just a note that this text was and is in the Supplementary Information. We now write, “Since *Ae. bicornis* and *Ae. speltoides* are species complexes, there may be variation in the presence/absence of *YrNAM* homologs within these species.”

On the other hand, I also read the other reviewer's comments and the rebuttals by the authors. In the revised version, the authors state in response to Reviewer 1 that "Unexpectedly, PC1213 transgenic T2 plants showed resistance (ITs=2) to CYR34 (Fig. 3b), which were likely caused by pyramiding Yr10 and Yr6 in Fielder (ref 17) and Yr9 in CB037 (ref 18)".

If this 'masking' assumption is correct, then CYR34 should be avirulent on cv Fielder, however, strangely, CYR34 is virulent on cv Fielder (see Fig. 3B). This indicates that the transgenics are no longer race specific, possibly suggesting that they express a non-specific ectopic resistance, e.g. due to over-expression. This is actually quite an important point viz-a-viz the identity of the cloned gene and its relationship with Yr10. In conclusion, I am unable to reconcile this observation with YrNAM being Yr10. I feel this requires further consideration and look forward to see what Reviewer 1 has to say about this.

Response: We are sorry for the confusion from the sentence “which were likely caused by pyramiding *Yr10* and *Yr6* in Fielder¹⁷ and *Yr9* in CB037¹⁸”. We note that the CYR34 is somewhat virulent on wild-type Fielder but avirulent on transgenic Fielder

expressing *YrNAM*. The wild-type Fielder is moderately resistant with some hypersensitive response that could be enhanced by the addition of *YrNAM*. We revised the sentence to: “PC1213 transgenic T₂ plants were resistant (ITs=2) to CYR34 (Fig. 3b), which may have been caused by stacking *YrNAM* with *Yr6* in Fielder¹⁷ and *Yr9* in CB037¹⁸”. We also added a sentence, “The wild-type Fielder is moderately resistant to CYR34 as indicated by a modest hypersensitive response (HR) (Fig. 3b).”

YrNAM is race-specific in the tests using cv. Moro and AvoSYr10NIL; it appears to be race non-specific in the transgenic experiment. However, it is hard to tell whether transgenic *YrNAM* will provide race non-specific resistance by testing only one or a few races. Anyway, we no longer claim that *YrNAM* is *Yr10*, as indicated above in the response to the editor.

II. Reviewer #3:

My previous concerns/comments have been well addressed. Here are two minor comments for the revision:

L95: “However, CYR34 is virulent to Yr10 because it caused abundant sporulation in Moro, P8-13, and P10-46 (Fig. 3b). Unexpectedly, PC1213 transgenic T₂ plants showed resistance (ITs=2) to CYR34 (Fig. 3b), which were likely caused by pyramiding Yr10 and Yr6 in Fielder¹⁷ and Yr9 in CB037¹⁸.” In the revision, the authors hypothesized that Fielder carrying Yr6 caused the resistance to CYR34. Since Fielder was used as the control, why not present data of the controls to support the hypothesis?

Response: Thanks for the question and we are sorry that we didn’t make it clear. In fact, as we responded to Reviewer #2, the wild-type Fielder has moderate resistance to CYR34, which may be due to *Yr6* or other *Yr* genes; this could be enhanced to high resistance (without any sporulation) by the addition of *YrNAM*. The phenotype for the control of Fielder is presented in Fig. 3b, and is now explicitly mentioned, as indicated above.

L151: What is localized expression in “which might be due to highly localized expression”?

Response: Sorry for the confusion. We deleted the sentence.

III. Reviewer #4:

There remain two questions should be addressed to identify the candidate gene as Yr10.

Firstly, according to the gene-for-gene theory, the race-specific resistance genes containing Yr10 will initiate plant disease resistance response only when they recognize the corresponding avirulent genes. Results in this study showed that YrNAM transgenic wheat lines were high resistant to all the tested Pst races comprise of CYR29, CYR31, CYR32, CYR33 and CYR34 as well as a mixture of spores from naturally infected wheat fields, thus, which could not be defined as Yr10. Secondly, Yr10 was identified and named a long time ago using different Pst races, and the races used in this study are different from those at that time. Therefore, to define the real Yr10 also needs to choose the same Pst races to verify.

Response: As indicated above in the response to the editor, we no longer claim that *YrNAM* is *Yr10*.

We greatly appreciate all the efforts of the editor and the reviewers on our manuscript!